# Multi-Ion-Based Modelling and Experimental Investigations on Consistent and High-Throughput Generation of a Micro Cavity Array by Mask Electrolyte Jet Machining

**DOI:** 10.3390/mi13122165

**Published:** 2022-12-07

**Authors:** Ming Wu, Zhongning Guo, Jun Qian, Dominiek Reynaerts

**Affiliations:** 1Department of Mechanical Engineering, Katholieke Universiteit Leuven, Oude Markt 13, 3000 Leuven, Belgium; 2Department of Computer Science, Katholieke Universiteit Leuven, Oude Markt 13, 3000 Leuven, Belgium; 3Members Flanders Make, 3000 Leuven, Belgium; 4College of Mechanical and Electrical Engineering, Guangdong University of Technology, Guangzhou 510006, China

**Keywords:** micro surface structures, mask electrolyte jet machining, electrochemical micro machining

## Abstract

The controllability and consistency in the fabrication of micro-textures on large-scale remains a challenge for existing production processes. Mask electrolyte jet machining (MEJM) is an alternative to Jet-ECM for controllable and high-throughput surface microfabrication with more consistency of dimensional tolerances. This hybrid configuration combines the high-throughput of masked-ECM and the adjustable flow-field of jet-ECM. In this work, a duckbill jet nozzle was introduced to make MEJM more capable of batch micro-structuring. A multiphysics model was built to simulate the distribution of electrochemical reaction ions, the current density distribution, and the evolution of the shape of the machined cavity. Experimental investigations are presented showing the influence of the machining voltage and nozzle moving speed on the micro cavity. Several 35×35 micro cavity arrays with a diameter of 11.73–24.92 μm and depth of 7.24–15.86 μm are generated on 304 stainless steel.

## 1. Introduction

Micro cavity arrays, as a typical surface microstructure, are broadly applied for heat exchangers [1,2], friction and wear [3], anti-fouling [4,5], etc. Recently, it has been reported that cutting tools [6,7] textured with micro cavities could reduce the cutting force, average friction coefficient, and cutting tool wear, which is useful for machining difficult-to-machine materials. Currently, several technologies have been introduced to manufacture micro cavities on metallic surfaces, such as femtosecond laser micromachining [8], micro-milling [9] and micro-electrical discharge machining [10].

Compared to the aforementioned methods, electrochemical micromachining (EMM) [11] is a promising method for preparing micro cavities [12], as it has unique advantages such as a good control on cavity profile, the potential for multi-response optimization [13], independence of material hardness [14] and toughness [15], absence of a heat-affected layer, lack of process related tool wear and burrs, and a high-throughput capability. Through-mask electrochemical micromachining (TMEMM) is a promising method for generating array-like surface microstructures. In this method, the workpiece surface is covered by a patterned mask, and the machining region is exposed. Subsequent electrochemical micromachining dissolves the exposed area to create the surface texture. With this method, several kinds of surface textures can be prepared, such as micro cavity arrays and micro groove arrays [16]. Wang et al. [17] reported fabrication of a micro cavity array with a diameter of 40 μm on a metallic cylindrical surface by using TMEMM. In the work of Qu et al. [18], a dry-film photoresist was used as a mask during through-mask electrochemical micromachining to successfully fabricate micro cavity arrays (each cavity about 94 μm in diameter and 22.7 μm deep) on inner cylindrical surfaces. Qu [19] proposed a modified micro-scale pattern transfer without involving photolithography of substrates. In their work, a through movable dry-film mask electrochemical micromachining was employed for fabrication of micro cavities of 109.4 μm in diameter and 15.1 μm in depth.

Besides the aforementioned masked-ECM methods, Jet electrochemical machining (Jet-ECM) has also been shown to be an effective approach for generating deep micro cavities [20]. The unique characteristic of this technology is that the electrolyte is ejected from the metallic nozzle to the workpiece with high velocity, which is helpful for preparing deep micro cavities as the electrolyte in the micro cavity can be renewed rapidly [21]. Jet-ECM has been used to fabricate micro-grooves and micro cavities, showing that it is a flexible method [22]. Hackert et al. [23] employed Jet-ECM for generating micro cavities by using a metallic nozzle with an inner diameter of 100 μm. As the depth increased from 37 μm to 90 μm, the diameter of the micro cavity was enlarged from 173 μm to 220 μm, and the machining localization was reduced. Because the workpiece surface is exposed to jet without side insulation, it often leads to undercutting and stray corrosion at the edge of the micro cavity, and the phenomenon is worsening with an increase in depth, which reduces the machining accuracy and surface quality. Yan [24] presented a reciprocating electrolyte jet machining technology with prefabricated mask (REJP) which was used to generate a circular cavity array of about 43 μm in depth and 822 μm in diameter on a cast-iron cylinder liner.

These aforementioned methods exhibit a rather low machining efficiency and do not meet the requirements of mass-fabrication of micro cavity like micro-structures. For efficient electrochemical machining of micro cavity array structures, enhanced electrolyte renewal can be helpful for machining high aspect ratio cavities, and reducing the undercutting can improve the machining localization. Mask electrolyte jet machining (MEJM) is an alternative to Jet-ECM/TMEMM for surface microfabrication with more consistency of dimensional variation [25,26]. MEJM is a hybrid configuration which combines the advantages of through-mask electrochemical machining, which is a high-throughput process, and of jet electrochemical machining, with its adjustable flow field.

In the present work, a duckbill jet nozzle is introduced to make MEJM more capable of batch micro-structuring. A multiphysics model is developed to simulate the electric field distribution and micro cavity forming process of the electrolyte. Experimental investigations regarding the influences of machining voltage and nozzle moving speed on the micro cavity are presented. Optimization of the experimental parameters is performed. Finally, the efficient machining of a large number of micro cavities on a stainless-steel plate is demonstrated.

## 2. Process Principle and Theoretical Analysis

### 2.1. Process Principle

The schematic view of MEJM using a duckbill nozzle is shown in Figure 1a. During the machining process, the high-speed electrolyte is sprayed from the metallic nozzle and the nozzle scans on the workpiece. Meanwhile, the high-speed electrolyte flow reaches the exposed workpiece through the micro holes in the mask. Finally, a micro cavity array can be generated when a sufficient voltage is applied between the metallic nozzle and the workpiece. More specifically, a metallic nozzle was employed to provide a stable and high-speed jet flow for workpiece and the renewal of electrolyte in the small machining area, which was useful for generating deep micro cavities. This method is highly flexible and enables machining of large areas.

A finite element model (FEM) is developed to investigate the electric field and current density distribution on the workpiece. The profile evolution of the micro cavities can be predicted by this FEM. A 2D diagram of this process configuration is shown in Figure 1b, the geometric and simulation parameters of the model are listed in Table 1.

During machining, the electrolytic products and joule heat will be rapidly removed from the machining area by a high velocity flow of electrolyte. Therefore, the heat effects in the electrochemical reactions do not need to be considered in this case. One of the most significant features of MEJM is the electrolyte flow direction that is changing over time, which refers to a changing concentration gradient in the bulk electrolyte. The current density $\vec{J}$ in the electrochemical cell can be represented by the ion transportation:(1)$$\vec{J}=F\sum z_{i}\vec{N_{i}}$$
where $z_{i}$ is the valence for species *i*, and $\vec{N_{i}}$ is the flux of ions which is the result from:

1Diffusion:${\vec{N}}_{D}$
(2)$$N_{D}=-D_{i}\nabla c_{i}$$
where $D_{i}$ is the diffusion coefficient, $c_{i}$ the concentration. Here, H^+^, OH^-^, Na^+^, NO_3_ and Fe^3+^ were the ions taking part in the electrochemical reactions, the diffusion coefficients are shown in Table 2.2Convection:${\vec{N}}_{C}$
(3)$$N_{C}=\vec{u}c_{i}$$
where $\vec{u}$ is the velocity field. In this case, the electrolyte flow rate is considered as laminar flow and can hence be represented by the Navier–Stokes equations:
(4)$$\left\{\begin{array}{l} \rho \frac{\partial \vec{u}}{\partial t}+\rho \left(\vec{u}\cdot \nabla \right)\vec{u}=-\nabla p+\mu \Delta \vec{u}+\rho \vec{g}\\ \frac{\partial \rho }{\partial t}+\rho \nabla \vec{u}=0 \end{array}\right.$$3Electric migration:
${\vec{N}}_{E}$
(5)$$N_{E}=-\frac{FD_{i}z_{i}ci}{RT}\nabla \phi$$
where *F* is the Faraday constant, *R* the ideal gas constant, *T* the temperature and ϕ the electric potential in the interelectrode gap, which can be described by Laplace’s equation with a number of certain boundary conditions:
(6)$$\left\{\begin{array}{cc} \nabla^{2}\phi & =\frac{\partial^{2}\phi }{\partial x^{2}}+\frac{\partial^{2}\phi }{\partial y^{2}}=0\\ \phi {|}_{\Gamma_{1}} & =0\;V\;(\mathrm{Cathode}\;\mathrm{boundaries})\\ \phi {|}_{\Gamma_{2}} & =10\;V\;(\mathrm{Anode}\;\mathrm{boundaries})\\ \frac{\partial \phi }{\partial \vec{n}}{|}_{\Gamma_{5,6}} & =0\;V\;(\mathrm{Insulation}\;\mathrm{boundaries}) \end{array}\right.$$
where $\vec{n}$ is the normal phase vector of the boundary.

Therefore, the flux ${\vec{N}}_{i}$ of species in the electrolyte can be given by Equation (Equation 7):(7)$$\begin{array}{cc} {\vec{N}}_{i} & ={\vec{N}}_{D}+{\vec{N}}_{C}+{\vec{N}}_{E}\\ & =-D_{i}\nabla c_{i}+\vec{u}c_{i}-\frac{FD_{i}z_{i}ci}{RT}\nabla \phi \end{array}$$

The boundary conditions for the ion transportation are as follows:

Inflow(inlet):(8)$$\left\{\begin{array}{cc} \vec{u}={\vec{u}}_{0} & \\ c_{i}=c_{i,0} \end{array}\right.$$Outflow(outlet):(9)$$\left\{\begin{array}{ll} p=0 & \\ \vec{n}\cdot D_{i}\nabla c_{i}=0 \end{array}\right.$$Open boundary:
− As no viscous stress is set for the laminar flow, it does not impose any constraint on the pressure:(10)$$\mu \left(\nabla \vec{u}+{\left(\nabla \vec{u}\right)}^{T}\right)\cdot \vec{n}=0$$− For the ion transportation:(11)$$\left\{\begin{array}{ll} -\vec{n}\cdot {\vec{N}}_{i}=0, & \vec{n}\cdot \vec{u}⩾0\\ c_{i}=c_{i,0}, & \vec{n}\cdot \vec{u}<0 \end{array}\right.$$Anode:(12)$$\mathrm{Fe}+2H_{2}O\to {\mathrm{Fe}}^{3+}+O_{2}+4H^{+}+7e^{-}$$Cathode:(13)$$2H_{2}O+2e^{-}\to H_{2}+2OH^{-}$$

According to Faraday’s law, the normal dissolution velocity $\vec{v_{n}}$ on the anode boundary can be given by:(14)$${\vec{v}}_{n}=\eta \frac{M}{zF}\vec{J}$$
where *M* is the molar mass of the workpiece material, η is the coefficient of material removal efficiency weighted by the pulse current and is set to 62.56% in this model.

By solving Equations (Equation 2), (Equation 3), (Equation 5), (Equation 6) and (Equation 14), the electric field, current density distribution and the material removal process can be calculated.

To make the model applicable, the following simplifying assumptions are made.

The deformation of the workpiece domain and subsequently the moving boundary is often addressed using the Arbitrary Lagrangian–Eulerian (ALE) finite element technique, in which the “elements” may be modified but cannot be produced or destroyed during the simulation process. To describe the workpiece material removal process that occurs between the workpiece and photoresist mask, however, there must be an interface where the workpiece and photoresist mask may deform in their respective directions, which makes the creation of a gap between them essential. In this work, a virtual gap with 0.1 μm is set between mask and workpiece to ensure the anode boundary to move properly.Theoretically, the fabricated micro features will influence the shape of the moving electrolyte jet column. However, it is assumed that, in practice, because of the low moving speed of the nozzle (1–4 mm/s) and the micrometer scale of the features generated, the electrolyte column remains unchanged across the machining process. The bulk electrolyte layer over the workpiece is assumed to be constant because the electrolyte flow pressure is too low (20 kPa) to cause a hydraulic jump associated with the flow pressure used in conventional Jet-ECM (500 kPa). The slit width of the duckbill nozzle in this work is the same as the diameter of the cylindrical nozzle used in Ref. [25], and the IEG is likewise the same. Therefore, the geometry shape of the electrolyte domain from Ref. [25] is used to develope the multiphysics model in this work.

The numerical simulation model is built as shown in Figure 1b,c. The model was built using a free triangular mesh, and the deformed region was refined to improve the calculation accuracy. In this work, the numerical simulations were performed by COMSOL^®^ Multiphysics software.

### 2.2. Simulation Results

The profile evolution and corresponding distribution of current density norm, which is the absolute magnitude of the current density vector, on the reaction interface can be seen in Figure 2, the diameter and depth of the machined cavity increased as the electrochemical reaction progressed. Because of the “edge effect” in the electric field, at *t* = 0 s, the current density norm was slightly higher in the boundary between the photoresist and workpiece than that in the center of micro cavity.

Later in the process, the distribution is inverted, the current density is then always higher in the center than that in the boundary. This will lead to a concave-like profile, i.e., a micro cavity.

The detailed simulation results at different machining times (*t* = 0 s: Figure A1; *t* = 1 s: Figure A2; *t* = 2 s: Figure A3; *t* = 3 s: Figure A4) are provided in Appendix A. As the electrochemical dissolution reaction progressed, the depth of micro cavity increased, causing an increasing distance between the cathode and the anodic area on the workpiece. In the meantime, the distance between the reaction interface and the nozzle, i.e., the cathode, is first reduced and then becomes longer again, the shortest distance between them was at the nozzle moving right above the reaction interface. In the presented simulation, the nozzle was right above the reaction interface at *t* = 2 s, as shown in Figure A3, and the current density norm reached its peak value and then was reduced.

As shown in Figure 3, the moving nozzle also changes the electrolyte flow direction both around and inside the machined cavity. The variation of the flow field and electric field will cause a variation of the ion distribution. According to Equation (Equation 5), positive ions will be repelled and negative ions will be attracted to the reaction interface since it is located at the surface of anode. Significantly more OH^-^ than H^+^ around the reaction area, which refers to electrochemical reactions occur under an extreme alkaline environment. Similarly, more NO_3_^-^ than Na^+^ ions were present in the reaction area. Due to the water depletion in Equation (Equation 12), H^+^ ions are produced at the anode surface and contribute to a rising trend of H^+^ concentration around the machining area. However, the electric migration caused by the potential gradient not only cancels this increasing concentration trend, but also inverts it into a lowering concentration trend; the concentration of H^+^ around the machined cavity is even lower than that of the bulk electrolyte. Since no Na^+^ ions are created at the anode surface, the concentration of Na^+^ around the machined cavity is even lower than the concentration of H^+^. The produced Fe^3+^ ions were also expelled from the reaction interface. In the meantime, the neutral byproducts, such as Fe(NO_3_)_3_ cannot be removed from the reaction area by the electric migration effect. These neural byproducts can be driven out by convection and diffusion. Here, the varying flow field makes the byproducts less prone to accumulate, which is what MEJM envisages.

Pulsed power supplies can reduce stray current corrosion and provide more precise machining than DC power supplies. Due to time step limitations, however, it is impractical to incorporate high-frequency pulsed electric current (2 kHz in the present experimental studies) into a multi-physical model, the cost of computation will be excessive. As a result, a simplification in practice is to build the ECM model with a coefficient (η in Equation (Equation 14)) to calculate the material removal rate rather than taking the pulse current into account at each time step [27]. In Figure 4, the profiles of the cavity from simulation and experiment are shown; the experimental processing parameters are identical to those of the simulations, with the exception of the pulse current. For additional information, please refer to Section 3.1) are presented. The shape and depth of the features from the simulation demonstrated a good agreement with that from the experiments. However, the simulated and experimental micro cavity diameters are 12 μm and 16.26 μm, respectively. This indicates that the depth of the micro cavity for experiments is higher than the simulation result. This may be because of the absence of pulsed current in our simulation, which can increase the localization of electrochemical reactions. In this study, a coefficient of material removal efficiency weighted by the pulse current is employed to fine-tune the simulation to match the results. However, only a linear correction is made for the material removal rate using this method. For instance, if the coefficient is increased in this study, the calculated cavity depth will match the real data, but the simulated diameter will expand. Thus, the simulated undercutting is larger than the experimental value. This indicates that a coefficient weighted by the pulse current is insufficient to capture the improvement in machining accuracy driven by the pulse current. Further work will focus on modelling of pulsed current and its effects on the electrical double layer.

## 3. Experimental Studies

### 3.1. Materials and Metrics

The workpiece material for experimental investigations was 304 stainless steel, and it was polished to mirror-surface levels (surface roughness <0.8 μm) to have uniform contact with the mask. The lithographic mask array holes are 5 μm in diameter. These micro-holes are distributed as a 35×35 square array, and the hole center-distance is 50 μm. The jet nozzle is implemented as a duckbill shaped nozzle with a slit length of 16 mm and a slit width of 2 mm. The experimental parameters used are listed in Table 3.

The surface topography of the micro cavity structure was obtained with a scanning electron microscope (S-3400N(II)). The size of micro cavity was measured by a laser scanning confocal microscope (Olympus OLS-4100). From the upper left corner to the lower right corner of the workpiece, 35 positions along the diagonal were uniformly selected for measurements. The diameter (*D*) and depth (*H*) of micro cavities were measured. The aspect ratio AR Equation (Equation 15), the etch factor EF Equation (Equation 16), and the standard deviation of diameter and depth $std_{D}$ Equation (Equation 17) and $std_{H}$ Equation (Equation 18) were calculated.
(15)$$AR=\frac{H_{i}}{D_{i}}$$
(16)$$EF=\frac{2H_{i}}{D_{i}-D_{0}}$$
(17)$$\left\{\begin{array}{ll} S_{W}=\sqrt{\frac{1}{N-1}\sum_{i=1}^{N}\left(D_{i}-\bar{D}\right)} & \\ \bar{D}=\frac{1}{N}\sum_{i=1}^{N}D_{i} \end{array}\right.$$
(18)$$\left\{\begin{array}{ll} S_{W}=\sqrt{\frac{1}{N-1}\sum_{i=1}^{N}\left(H_{i}-\bar{H}\right)} & \\ \bar{H}=\frac{1}{N}\sum_{i=1}^{N}H_{i} \end{array}\right.$$
where *N* the total number of measurements, $D_{0}$ is the diameter of cavities in the mask, $D_{i}$ the diameter of the *i*-th measured machined cavities *D*, $H_{i}$ the depth of *i*-th measured machined cavities.

### 3.2. Results and Discussion

#### 3.2.1. Influence of Applied Voltage

Voltages ranging from 10 V to 40 V were applied to investigate their effect on shape accuracy of micro cavity fabrication. A typical profile of a micro cavity array with good shape consistency (the standard deviation for diameter and depth are $std_{D}$ < 0.8 μm and $std_{H}$ < 0.4 μm, respectively) and single micro cavity at different applied voltages (10 V and 40 V) are shown in Figure 5.

The dimensions of micro cavities generated with different applied voltage, with a pulse duty cycle of 50%, pulse frequency of 1kHz, and a nozzle moving speed of 1mm/s are shown in Figure 6 and Table 4. The violin plot, as shown in Figure 6, is presented for a better understanding of the processing pattern. The diamond-shaped areas generated by the multivariate KDE-based probabilistic density function are used to construct the violin plot, which examines the distribution of the results with different processing parameters. This data visualization method provides a statistical and intuitive way of interpreting the data. For more details, please refer Figure A5 in Appendix B.

It can be seen that the diameter of micro cavities increased from 18.47±0.56μm (standard deviation) to 23.29±0.74μm as the voltage increased from 10V to 40V. Furthermore, the depth increased from 8.08±0.02μm to 11.04±0.39μm with an increase in voltage, thereby showing a similar trend as for the diameter. With an increase in voltage, the current density becomes higher. Therefore, the amount of material removal in a given machining time also increased. As a result, the diameter and depth of the micro cavities gradually increased, and the removal rate follows the general law of Faraday dissolution as shown in Equation (Equation 14).

The values of the etch factor (EF) and the aspect ratio (AR) of the micro cavity at different voltages are shown in Figure 6 and Table 5. As the voltage increased, the value of EF is around 1.202 and the aspect ratio of micro cavity ranges from 0.438 to 0.475. This shows that the presented MEJM process generates an aspect ratio that is higher than others described in the literature.

It is worth mentioning that the standard deviations of diameter and depth slightly increased as the voltage increased. This indicates that the machining stability was reduced as the amount of removed material increased.

#### 3.2.2. Influence of Nozzle Moving Speed

A nozzle moving speed ranging from 1mm/s to 4mm/s was employed to investigate their effect on the shape accuracy of micro cavity fabrication. The typical profile of a micro cavity array with good shape consistency and single micro cavity at a different nozzle moving speed (2mm/s and 4mm/s) are shown in Figure 7.

The diameter and depth of the micro cavities generated at different nozzle moving speeds with an applied voltage of 30V, a pulse duty cycle of 50%, and a pulse frequency of 2 kHz are shown in Figure 8 and Table 6. It can be seen that the diameter of the micro cavities decreased from 22.07±0.71μm to 17.21±0.52μm with an increase in nozzle moving speed from 1mm/s to 4mm/s. Furthermore, the depth decreases from 10.16±0.31μm to 7.70±0.20μm with an increase in nozzle moving speed. As the nozzle moving speed increases, it is equivalent to reducing the processing time, and thus the diameter and depth of the micro cavities have been reduced.

The values of EF and AR of the micro cavities using different nozzle moving speeds are shown in Figure 8 and Table 7. Even at different nozzle moving speeds, the value of EF is around 1.2, showing the same trend as that for the voltage. The aspect ratio of the micro cavities ranges from 0.461 to 0.448. In this case, the EF of this MEJM technology is comparable to those observed in the literatures. Qu et al. [18] used a 50μm thick dry-film photoresist as a mask during the TMEMM process to fabricate a micro cavity with an EF of 1.03 and 94μm in diameter on a cylindrical inner surface, when the original mask aperture is 50μm in diameter.

The standard deviations of diameter and depth were slightly decreased ($std_{D}$: from 0.71 to 0.52, $std_{H}$) as the nozzle moving speed increased. This shows that the machining reliability was slightly improved as the amount of removed material decreased. This is probably because the deceased total machining time will decrease the risk of mask failure provoked by electrolyte flow flush.

## 4. Conclusions

In this study, a method of mask-based electrolyte jet machining with a duckbill nozzle was proposed for mass fabrication of a micron-sized micro cavity array on the surface of metal parts. The micro cavity shape evolution process and electric-field/current density distribution were simulated using the COMSOL^®^® Multiphysics software. A micro cavity array structure with a good shape consistency (the standard deviation for machined cavity diameter and depth can be as small as 0.52 and 0.2, respectively) was fabricated by MEJM. The design of experiments explored the influence of applied voltage and nozzle moving speed on the size and topography of the micro cavities. It can be concluded that the simulations showed good agreement with the experimental profile. However, there was less machining depth in simulated profiles, possibly because of the absence of considering a pulsed current.

It was observed in experiments that with an increase in process-voltage, the dimensions of micro cavities and depth-to-diameter ratios gradually increased. The diameter and depth increased by 26.08% and 36.57%, respectively, as the voltage increased from 10V to 40V. When the processing voltage is 40V, the size of the micro cavities is highest. At low voltages, the micro cavities are shallow. Therefore, preference should be given to medium voltage. When the applied voltage is 20V, the EF and AR values are the highest.

With an increase in the nozzle moving speed, the diameter and depth of the micro cavities decrease. The diameter and depth decreased by 22.04% and 24.23%, respectively, as the nozzle moving speed increased from 1mm/s to 4mm/s.

Overall, micro cavity structures were successfully fabricated using the proposed MEJM technology and a duckbill nozzle. This work is an initial step towards using MEJM technology for deterministic and efficient fabrication of micro-structures such as cavity/grooves on large workpieces. The technology can be further changed to textured, curved and free-form surfaces using flexible masks. The technology has potential applications in the texturing of bearings for improved lubrication, mimicking of artificial bearing defects and micro-structuring of mould surfaces for improved ceramic/polymer injection molding. The machining precision is mainly dependent on the resolution of the lithographic mask, rather than the size of the tool; limitations on tool dimensions are overcome with this technology. The holes in the masks can be fabricated down to a nanometer scale using state-of-the-art lithography techniques. With improved electrolyte recycling systems, this technology can be a cost effective technology as it does not involve capital costs such as those of a laser texturing process using femto-second lasers, expensive optics and beam manipulation peripherals.

## Figures and Tables

**Figure 1 micromachines-13-02165-f001:**
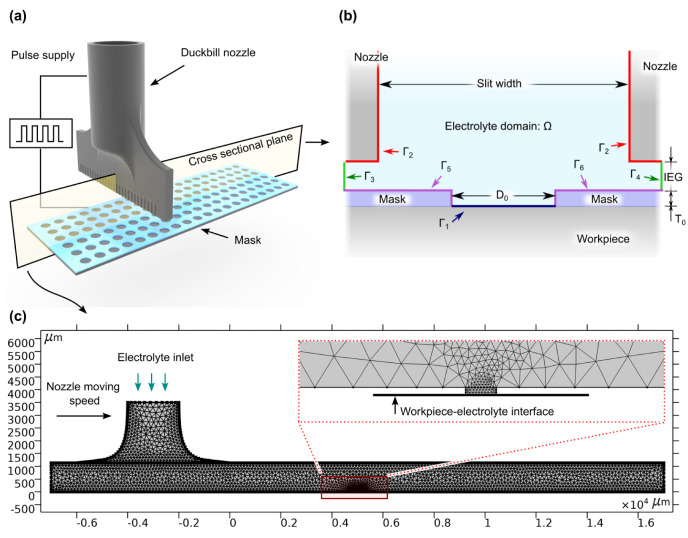
(**a**) MEJM process schematic view (not to scale). In this work, the lithographic mask array holes are 5μm in diameter. The center−to−center distance of these micro−holes is 50μm, and they are distributed as several 35×35 square arrays; (**b**) The 2D model diagram of MEJM (not to scale); (**c**) FEM simulation geometry and mesh.

**Figure 2 micromachines-13-02165-f002:**
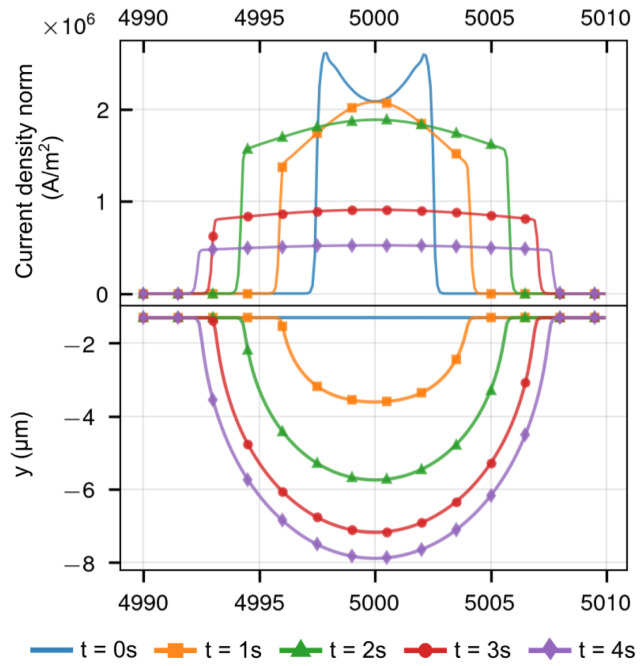
The profile evolution and corresponding current density norm on the reaction interface.

**Figure 3 micromachines-13-02165-f003:**
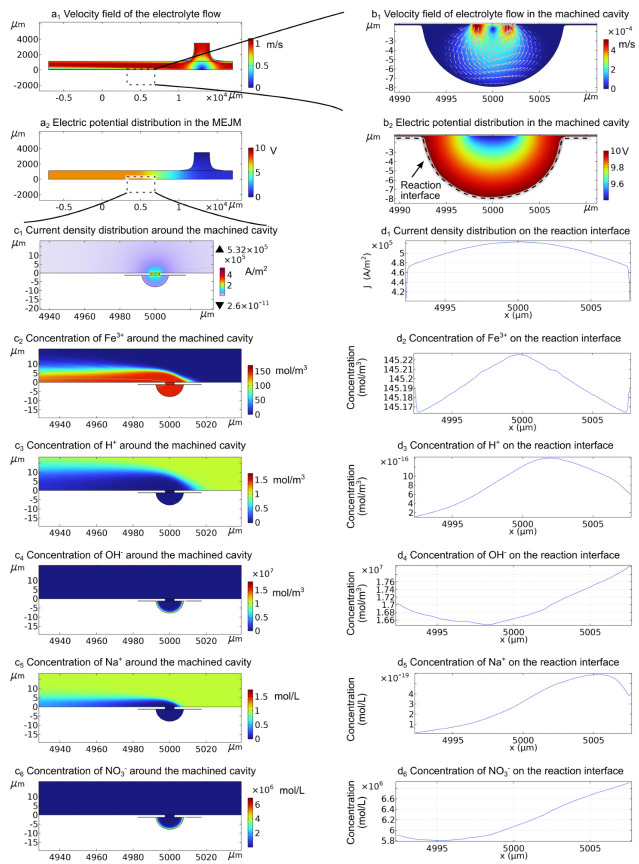
Simulation results at *t* = 4 s. ($a_{1}$): velocity field of the electrolyte flow; ($b_{1}$): velocity field of electrolyte flow in the machined cavity; ($a_{2}$) electric potential distribution in the MEJM; ($b_{2}$) electric potential distribution (V) in the machined cavity; ($c_{1}$) normal current density distribution around the machined cavity; ($d_{1}$): normal current density distribution on the reaction interface; ($c_{2}$–$c_{6}$): concentration $\left(\mathrm{mol}/m^{3}\right)$ of Fe^3+^, H^+^, OH^-^, Na^+^, and NO_3_^-^ around the machined cavity; ($d_{2}$–$d_{6}$): concentration ($\mathrm{mol}/m^{3}$) of Fe^3+^, H^+^, OH^-^, Na^+^, and NO_3_^-^ on the reaction interface.

**Figure 4 micromachines-13-02165-f004:**
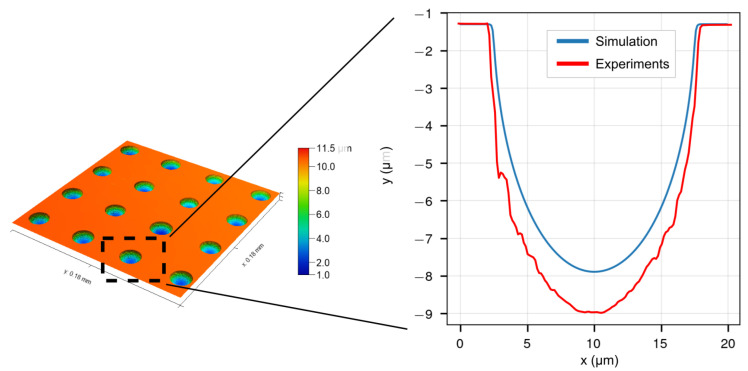
The cross sectional profile of cavities from simulation and experiments.

**Figure 5 micromachines-13-02165-f005:**
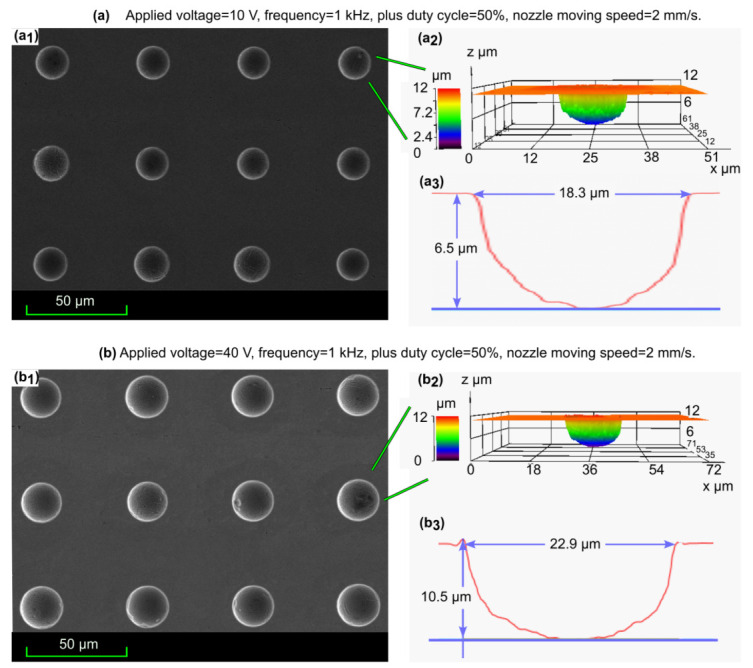
The typical profile of micro cavities at different applied voltage: (**a**): 10V; (**b**): 40V; ($a_{1}$,$b_{1}$): SEM images of machined cavities; ($a_{2}$,$b_{2}$): Confocal laser scanning microscope images of machined cavities; ($a_{3}$,$b_{3}$): Cross-sectional images of corresponding machined cavities.

**Figure 6 micromachines-13-02165-f006:**
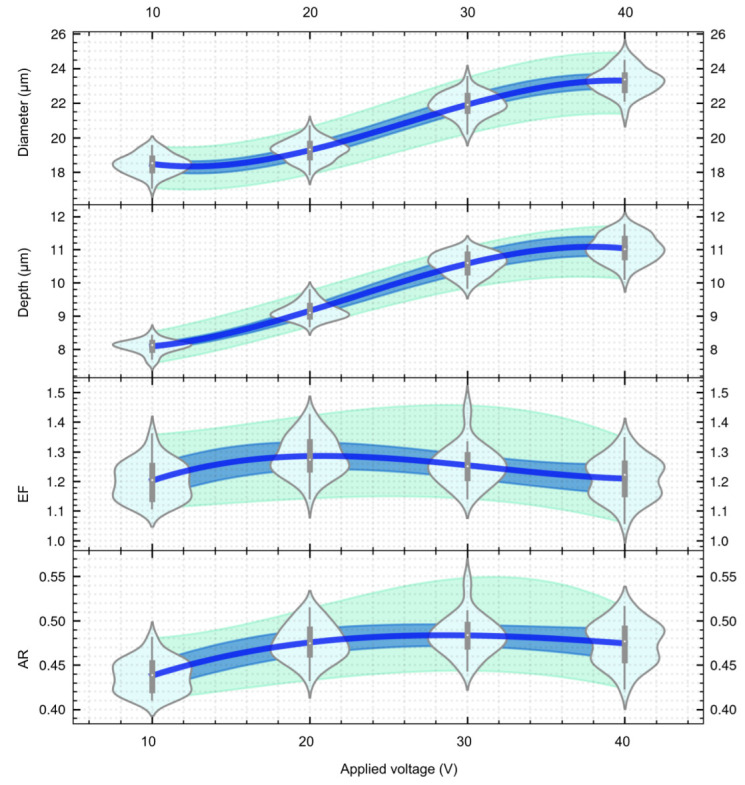
The effect of voltage on the dimensions of micro cavities.

**Figure 7 micromachines-13-02165-f007:**
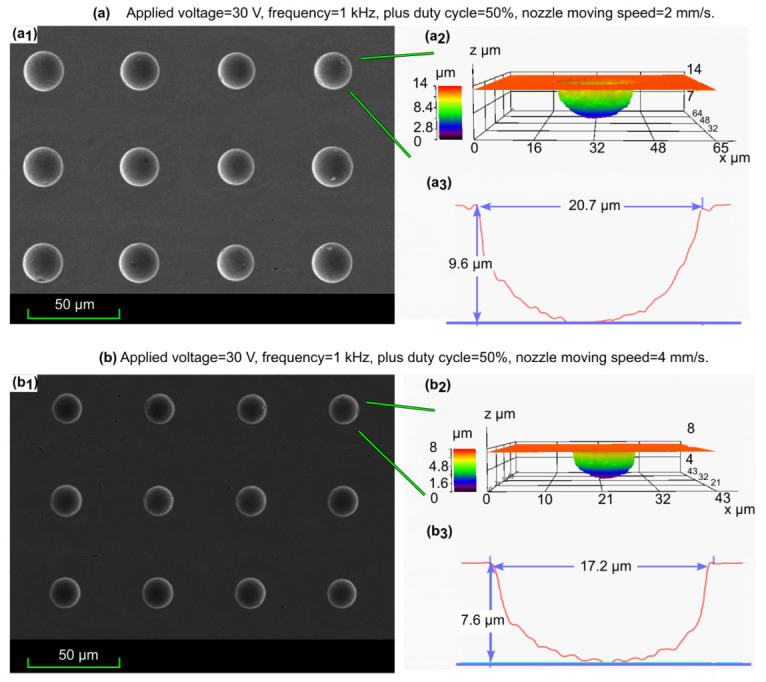
The typical profile of micro cavities at different nozzle moving speed: (**a**): 2mm/s; (**b**): 4mm/s; ($a_{1}$,$b_{1}$): SEM images of machined cavities; ($a_{2}$,$b_{2}$): Confocal laser scanning microscope images of machined cavities; ($a_{3}$,$b_{3}$): Cross-sectional images of corresponding machined cavities.

**Figure 8 micromachines-13-02165-f008:**
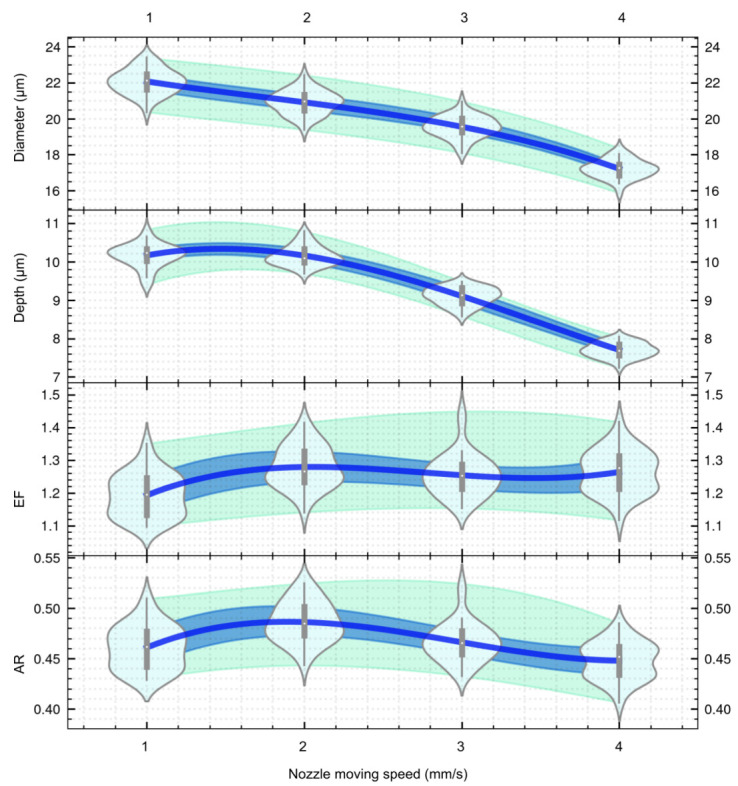
The effect of nozzle moving speed on the dimensions of micro cavities.

**Table 1 micromachines-13-02165-t001:** The parameters and initial conditions for the simulation.

Model Parameters	Value
Diameter of the dimple in the mask, $D_{0}$	5μm
Thickness of the mask, $T_{0}$	1.2μm
Inter-electrode gap, IEG	3.5mm
Duckbill nozzle slit length,	16mm
Duckbill nozzle slit width,	2mm
Density of electrolyte, ρ	$1100\;\mathrm{kg}/m^{3}$
Electrolyte temperature, *T*	298K
Electrolyte conductivity, σ	10S/m
Faraday constant, *F*	96,486C/mol
Applied voltage, *U*	10V
Nozzle moving speed, *v*	2mm/s
Molar gas constant, *R*	8.314472J/(K·mol)
Electrolyte pressure	20kPa
Initial concentration of Na^+^	1mol/L
Initial concentration of NO_3_^-^	1mol/L
Initial concentration of H^+^	0mol/L
Initial concentration of OH_3_^-^	0mol/L
Initial concentration of Fe^3+^	0mol/L

**Table 2 micromachines-13-02165-t002:** Diffusion coefficients at 293.15K.

Species *i*	Diffusion Coefficient $D_{i}$ ($10^{-9}\;m^{2}/s$)
Na^+^	1.33
NO_3_^-^	1.90
H^+^	9.31
OH^-^	5.26
Fe^3+^	1.24

**Table 3 micromachines-13-02165-t003:** Experimental parameters.

Parameters	Value
Applied voltage	10, 20, 30, 40V
Pulse frequency	2kHz
Pulse duty cycle	50%
Nozzle moving speed	1, 2, 3, 4mm/s
Inter-electrode-gap	3.5mm
Electrolyte concentration	10% (wt.%) aq. NaNO_3_
Electrolyte temperature	25 °C
Electrolyte pressure	20kPa
Diameter of cavitys in the mask	5μm
Mask thickness	1.2 μm
Workpiece material	Stainless steel 304

**Table 4 micromachines-13-02165-t004:** Diameter and depth of machined cavities with different applied voltage.

Voltage (V)	10		20		30		40	
	D	H	D	H	D	H	D	H
mean	18.47	8.08	19.27	9.15	21.90	10.58	23.29	11.04
std	0.56	0.20	0.60	0.26	0.69	0.32	0.74	0.39
min	17.11	7.58	17.88	8.68	20.23	9.85	21.37	10.12
25% †	18.11	7.99	18.88	8.98	21.55	10.31	22.77	10.78
50% ‡	18.52	8.13	19.33	9.10	21.92	10.61	23.38	11.02
75% ⋆	18.80	8.20	19.66	9.32	22.44	10.87	23.62	11.34
max	19.53	8.53	20.64	9.77	23.50	11.10	24.92	11.73

† 25th percentile; ‡ 50th percentile (median); ⋆ 75th percentile.

**Table 5 micromachines-13-02165-t005:** EF and AR value of machined cavities with different applied voltage.

Voltage (V)	10		20		30		40	
	EF	AR	EF	AR	EF	AR	EF	AR
mean	1.20	0.44	1.29	0.48	1.25	0.48	1.21	0.47
std	0.06	0.02	0.07	0.02	0.07	0.02	0.07	0.02
min	1.11	0.41	1.14	0.43	1.14	0.44	1.06	0.42
25% †	1.14	0.42	1.24	0.46	1.21	0.47	1.16	0.46
50% ‡	1.20	0.44	1.27	0.48	1.25	0.48	1.22	0.48
75% ⋆	1.25	0.45	1.33	0.49	1.29	0.50	1.26	0.49
max	1.36	0.48	1.42	0.51	1.46	0.55	1.35	0.52

† 25th percentile; ‡ 50th percentile (median); ⋆ 75th percentile.

**Table 6 micromachines-13-02165-t006:** Diameter and depth of machined cavities with different nozzle moving speed.

Speed (mm/s)	1		2		3		4	
	D	H	D	H	D	H	D	H
mean	22.07	10.16	20.91	10.16	19.55	9.10	17.21	7.70
std	0.71	0.31	0.66	0.26	0.60	0.23	0.52	0.20
min	20.37	9.40	19.38	9.68	18.08	8.58	15.86	7.24
25% †	21.63	10.02	20.48	9.99	19.24	8.92	16.84	7.57
50% ‡	22.13	10.24	20.98	10.10	19.56	9.13	17.27	7.69
75% ⋆	22.48	10.33	21.35	10.33	20.02	9.31	17.44	7.85
max	23.40	10.83	22.43	10.78	20.96	9.48	18.36	8.04

† 25th percentile; ‡ 50th percentile (median); ⋆ 75th percentile.

**Table 7 micromachines-13-02165-t007:** EF and AR of machined cavities with different nozzle moving speed.

Speed (mm/s)	1.00		2.00		3.00		4.00	
	EF	AR	EF	AR	EF	AR	EF	AR
mean	1.19	0.46	1.28	0.49	1.25	0.47	1.26	0.45
std	0.07	0.02	0.06	0.02	0.06	0.02	0.07	0.02
min	1.10	0.43	1.14	0.44	1.15	0.43	1.12	0.41
25% †	1.13	0.44	1.23	0.47	1.21	0.45	1.21	0.43
50% ‡	1.20	0.46	1.27	0.48	1.25	0.47	1.28	0.45
75% ⋆	1.25	0.48	1.33	0.50	1.29	0.48	1.31	0.46
max	1.35	0.51	1.41	0.52	1.45	0.52	1.42	0.48

† 25th percentile; ‡ 50th percentile (median); ⋆ 75th percentile.

## Data Availability

Not applicable.

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
