# Peer review of "Multi-Ion-Based Modelling and Experimental Investigations on Consistent and High-Throughput Generation of a Micro Cavity Array by Mask Electrolyte Jet Machining"

_micromachines, 2022, doi:10.3390/mi13122165_

Round 1
Reviewer 1 Report
Please refer to commented manuscript for detailed suggestions on revisions.

Author Response
We highly appreciate the reviewers’ insightful and helpful comments on our manuscript.
Many sentences of the manuscript have been carefully rewritten or reorganized to enhance the logic flow and make the statements stricter in a proper tone. We also would like to correct some typos/mistakes we made in the original manuscript.
Since most of the comments are in-line, and due to the space limits, we put all the revisions in-line. All the modifications along with the reviewer's comments can be found in the attached pdf file.

Reviewer 2 Report
Dear editor,
In my opinion this manuscript is original, clearly written with ideas and arguments that all are consistent with theoretical issues and practical considerations. I recommend this manuscript to be published after minor revision.
Author Response
We thank the reviewer for the kind words. We have revised our manuscript with some typos and format errors.

Reviewer 3 Report
Congratulations, I think this is a very nice paper.
I have only found some questions, that are probably just typos, but please check in the attahed PDF. These are in lines 92-93, 95-96, 97-98. I suppose the arrows on the top of the letters (vector symbol) should be at the centre of the letter, not shifted to the right as they are now. This is probably caused by centre alignment to the whole expression, containing a relatively wide lower index.
I think a correct arrow position should be as is in line 98, because the lower index i is narrow.
The same is even more pronounced in Eq. (7), (8) and (14), because these are shown next to each other.

Author Response
We thank the reviewer for pointing this out. The format errors have been revised.
In addition, some mistakes and formatting errors were corrected.

Round 2
Reviewer 1 Report
Please refer to commented manuscript for detailed suggestions on revisions.

Author Response
We have carefully addressed all the issues item by item as follows.
